# RE-EVALUATING WORD MOVER'S DISTANCE

## ABSTRACT

The word mover's distance (WMD) is a fundamental technique for measuring the similarity of two documents. As the crux of WMD, it can take advantage of the underlying geometry of the word space by employing an optimal transport formulation. The original study on WMD reported that WMD outperforms classical baselines such as bag-of-words (BOW) and TF-IDF by significant margins in various datasets. In this paper, we point out that the evaluation in the original study could be misleading. We re-evaluate the performances of WMD and the classical baselines and find that the classical baselines are competitive with WMD if we employ an appropriate preprocessing, i.e., L1 normalization. In addition, We introduce an analogy between WMD and L1-normalized BOW and find that not only the performance of WMD but also the distance values resemble those of BOW in high dimensional spaces.

## 1 INTRODUCTION

The optimal transport (OT) distance is an effective tool for comparing probabilistic distributions. Applications of OT include image processing (Ni et al., 2009; Rabin et al., 2011; De Goes et al., 2012), natural language processing (NLP) (Kusner et al., 2015; Rolet et al., 2016), biology (Schiebinger et al., 2019; Lozupone & Knight, 2005; Evans & Matsen, 2012), and generative models (Arjovsky et al., 2017; Salimans et al., 2018).

A prominent application of OT is the word mover's distance (WMD) (Kusner et al., 2015) for document comparison. WMD regards a document as a probabilistic distribution of words, defines the underlying word geometry using pre-trained word embeddings, and computes the distance using the optimal transport distance between two word distributions (i.e., documents). WMD is preferable because it takes the underlying geometry into account. For example, bag-of-words (BOW) will conclude that two documents are dissimilar if they have no common words, whereas WMD will determine that they are similar if the words are semantically similar (even if they are not exactly the same), as illustrated in Figure 1.

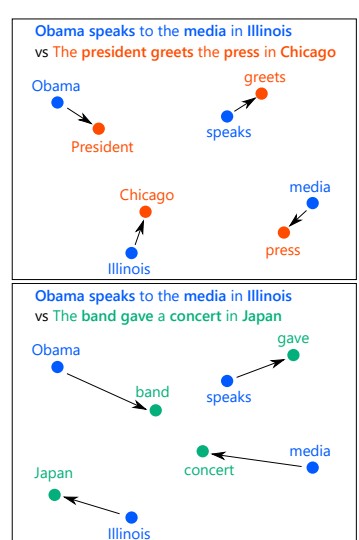

Figure 1: Neither pair of texts has common words. WMD can choose a similar sentence appropriately, whereas the BOW distance cannot distinguish these cases.

WMD has been widely used in NLP owing to this preferred property. For example, Kusner et al. (2015) and others (Huang et al., 2016; Li et al., 2019) used WMD for document classification, Wu et al. (2018) used WMD for computing document embeddings, Xu et al. (2018) used WMD for topic modeling, Kilickaya et al. (2017) and others (Clark et al., 2019; Zhao et al., 2019; 2020; Wang et al., 2020a; Gao et al., 2020; Lu et al., 2019; Chen et al., 2020b) used WMD for evaluating text generation. Many extensions have been proposed including supervised (Huang et al., 2016; Takezawa et al., 2021) and fast (Le et al., 2019; Backurs et al., 2020; Genevay et al., 2016; Dong et al., 2020; Sato et al., 2020b) variants. WMD is one of the fundamental tools used in NLP, and understanding the deep mechanism of WMD is crucial for further applications.

The most fundamental application of WMD is document classification. The original study on WMD (Kusner et al., 2015) conducted extensive experiments using kNN classifiers. Figure 2 shows the

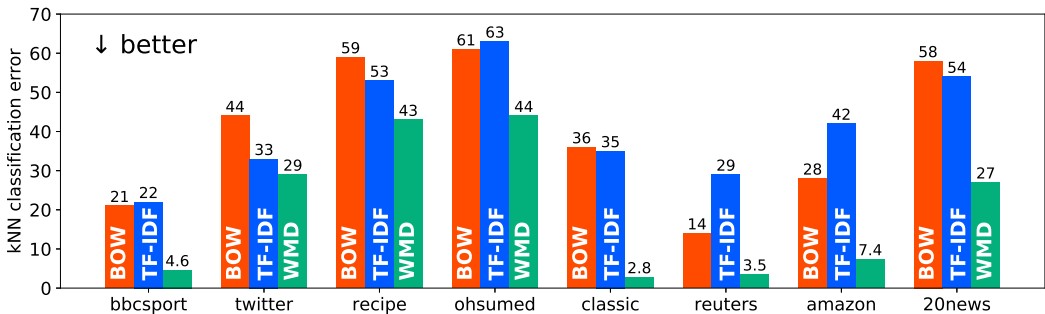

Figure 2: kNN classification errors reported in the original WMD paper (Figure 3 in (Kusner et al., 2015)). Lower is better. WMD outperformed the naive baselines by significant margins.

classification errors reported in (Kusner et al., 2015). This figure clearly shows that WMD is superior to classical baselines, BOW and TF-IDF[1]

Figure 2 is surprising in the following senses. First, WMD outperforms the classical baselines by excessively large margins. BOW and TF-IDF have long been recognized as effective tools for document classification. Although it is reasonable for WMD to outperform them, the improvements are surprisingly large. In particular, the performance is ten times better on the classic dataset and five times better on the bbcsport dataset. Such results are excessively impressive. Second, although TF-IDF is known to be more effective than BOW, it performs worse than BOW on the ohsumed, reuters, and amazon datasets. In fact, the number of misclassification doubles on the reuters datasets.

In this paper, we point out the possibility that the evaluations conducted in the original WMD study (Kusner et al., 2015) are misleading. Specifically, we found that the main improvements of WMD were due to normalization. Using the same normalization, WMD is comparable to BOW and TF-IDF, or WMD achieves improvements of only two to eight percent at the price of heavy computations. We also confirm that TF-IDF is more effective than raw BOW if we employ adequate normalization.

To understand the mechanism of WMD, we introduce an analogy between WMD and L1-normalized BOW. We experimentally find that the distribution of the distances between matched words is not Gaussian-like but two-modal in high dimensional spaces. We then find that not only the performance of WMD but also the distance values resemble those of BOW in high dimensional spaces.

The contributions of this paper are summarized as follows.

- We point out that the performance of WMD is not as high as we previously believed. The performance is comparable to classical baselines in document classification with the same normalization.

- We introduce an analogy between WMD and L1-normalized BOW (Proposition 1) and find that WMD resembles BOW in high dimensional spaces (Figure 5).

- We point out several confusing aspects in the evaluations conducted in the original study on WMD. We suspect that many readers and researchers are unaware of these issues. Clarifying them is crucial for a solid evaluation and analysis in this field

**Reproducibility.** We include our code in the supplementary materials. It contains a script to download datasets and pre-computed results, algorithm implementations, and evaluation code.

## 2 RELATED WORK

**Word Mover's Distance (WMD) and Optimal Transport (OT) in NLP.** WMD (Kusner et al., 2015) is one of the most thriving applications of OT. WMD can take the underlying word geometry into account and inherit many elegant theoretical properties from OT. The success of WMD has facilitated many applications of OT in NLP. EmbDist (Kobayashi et al., 2015) is a method concurrent

---

[1]It should be noted that (Kusner et al., 2015) used many stronger baselines such as LSI and LDA. We focus on BOW and TF-IDF because (i) BOW was used as the base performance (Figure 4 in (Kusner et al., 2015), Figure 4 in (Yurochkin et al., 2019)), and (ii) BOW is a special case of WMD (Propositoin 1).

with WMD. It also regards a document as a distribution of word embeddings but uses greedy matching instead of OT. Kumar et al. (2017) apply WMD to hidden representations of words instead of raw word embeddings. Yurochkin et al. (2019) consider a document as a distribution of topics and compute the document similarities using OT. They also use the OT distance for computing the ground distance of the topics. Alvarez-Melis et al. (2018) proposed a structured OT and applied it to a document comparison to take the positional consistency into account. Singh et al. (2020) consider a word as a probabilistic distribution of surrounding words and compute the similarity of the words using the OT distance of the distributions. Muzellec & Cuturi (2018) and others (Deudon, 2018; Frogner et al., 2019; Sun et al., 2018) embed words or sentences into distributions instead of vectors and compute the distance between embeddings using OT. Chen et al. (2019) and others (Li et al., 2020; Chen et al., 2020a) regularize the text generation models based on the OT distance between the generated texts and the ground truth texts to improve the generation. Nested Wasserstein (Zhang et al., 2020a) compares the distributions of sequences and is successfully used in imitation learning for text generation. Lei et al. (2019) use WMD to generate paraphrase texts for creating adversarial examples. Zhang et al. (2020b) introduced partial OT to drop meaningless words. Michel et al. (2017) use a Gromov Wasserstein-like distance instead of the standard OT to compare documents. Zhang et al. (2016) and others (Zhang et al., 2017b;a; Grave et al., 2019; Dou & Neubig, 2021) use OT to align word embeddings of different languages. Trapp et al. (2017) use WMD to compare compositional documents by weighting each document. Kilickaya et al. (2017) and others (Clark et al., 2019; Zhao et al., 2019; 2020; Wang et al., 2020a; Gao et al., 2020; Lu et al., 2019; Chen et al., 2020b) used WMD for evaluating text generation. BERTScore (Zhang et al., 2020c) is a relevant method, but it uses greedy matching instead of OT. To summarize, OT and WMD have been used in many NLP tasks. It is important to understand the underlying mechanism of WMD for further advancements in this field.

**Re-evaluation of Existing Methods.** Back in 2009, Armstrong et al. (2009) found that, although many studies have claimed statistically significant improvements against the baselines, most have employed excessively weak baselines, and the performance did not improve from the classical baselines in the information retrieval domain. Dacrema et al. (2019) recently found that many of the deep learning-based recommender systems are extremely difficult to reproduce, and for the methods whose results authors could reproduce, the performance was not as high as people had believed, and the deep approaches were actually comparable to classical baselines with an appropriate hyperparameter tuning. Their paper has had a large impact on the community and was awarded thr best paper prize at RecSys 2019. Similar observations have also been made in sentence embeddings (Arora et al., 2017; Shen et al., 2018), session-based recommendations (Ludewig et al., 2019), and graph neural networks (Errica et al., 2020) as well. In general, science communication suffers from publication and confirmation biases. The importance of reproducing existing experiments by third-party groups has been widely recognized in science (Lin, 2018; Sculley et al., 2018; Munafò et al., 2017; Collins & Tabak, 2014; Goodman et al., 2016).

## 3 BACKGROUNDS

### 3.1 PROBLEM FORMULATION

In this paper, we consider document classification tasks. Each document is represented by a bag-of-word vector $\boldsymbol{x} \in \mathbb{R}^m$, where $m$ is the number of unique words in the dataset. The $i$-th component of $\boldsymbol{x}$ represents the number of occurrences of the $i$-th word in the document. We focus on the kNN classification following the original paper. The kNN classification gathers $k$ samples of the smallest distances (with respect to a certain distance) to a test sample from the training dataset and classifies the sample to the majority class of the gathered labels. The design of the distance function is crucial for the performance of kNN classification.

### 3.2 WORD MOVER'S DISTANCE (WMD)

WMD provides an effective distance function utilizing pre-trained word embeddings. Let $\boldsymbol{z}_i$ be the embedding of the $i$-th word. To utilize OT, WMD first regards a document as a discrete probabilistic distribution of words by normalizing the bag-of-word vector:

$$\boldsymbol{n}_{\mathrm{L1}}(\boldsymbol{x}) = \boldsymbol{x} / \sum_i \boldsymbol{x}_i. \tag{1}$$

Table 1: Dataset statistics.

| | bbcsport | twitter | recipe | ohsumed | classic | reuters | amazon | 20news |
|---|---|---|---|---|---|---|---|---|
| Number of documents | 737 | 3108 | 4370 | 9152 | 7093 | 7674 | 8000 | 18821 |
| Number of training documents | 517 | 2176 | 3059 | 3999 | 4965 | 5485 | 5600 | 11293 |
| Number of test documents | 220 | 932 | 1311 | 5153 | 2128 | 2189 | 2400 | 7528 |
| Size of the vocabulary | 13243 | 6344 | 5708 | 31789 | 24277 | 22425 | 42063 | 29671 |
| Unique words in a document | 116.5 | 9.9 | 48.3 | 60.2 | 38.7 | 36.0 | 44.6 | 69.3 |
| Number of classes | 5 | 3 | 15 | 10 | 4 | 8 | 4 | 20 |
| Split type | five-fold | five-fold | five-fold | one-fold | five-fold | one-fold | five-fold | one-fold |
| Duplicate pairs | 15 | 976 | 48 | 1873 | 2588 | 143 | 159 | 59 |
| Duplicate samples | 30 | 474 | 66 | 3116 | 600 | 197 | 285 | 88 |

WMD defines the cost matrix $C \in \mathbb{R}^m \times \mathbb{R}^m$ as the distance of the embeddings, i.e., $C_{ij} = \|z_i - z_j\|_2$. The distance between the two documents $x$ and $x'$ is the optimal value of the following problem:

$$\underset{P \in \mathbb{R}^{m \times m}}{\text{minimize}} \quad \sum_{ij} C_{ij} P_{ij} \tag{2}$$

$$\text{s.t.} \quad P_{ij} \geq 0, P\mathbb{1} = n_{\text{L1}}(x), P^\top \mathbb{1} = n_{\text{L1}}(x'),$$

where $P^\top$ denotes the transpose of $P$, $\mathbb{1} \in \mathbb{R}^m$ is the vector of ones. Intuitively, $P_{ij}$ represents the amount of word $i$ that is transported to word $j$. WMD is defined as the minimum total distance to convert one document to another document. Let $\text{OT}(x, x', C) \in \mathbb{R}$ be the optimal value of eq. (2).

## 3.3 EXPERIMENTAL SETUPS

**Datasets.** We use the same datasets (Greene & Cunningham, 2006; Sanders, 2011; Joachims, 1998; Sebastiani, 2002; Lang, 1995) as in the original paper (Kusner et al., 2015) and use the same train/test splits as in the original paper. Three of the eight datasets have the standard train/test splits (e.g., based on timestamps), and the other datasets do not. Thus, the original paper used five random splits for such datasets. We refer to the former type as one-fold datasets and the latter as five-fold datasets. Table 1 shows the statistics. Here, we point out the first misleading point.

**Misleading Point 1** Many duplicate samples exist in the datasets.

The last two rows in Table 1 report the numbers of duplicate samples. Some of these samples cross the train/test splits, and some of them have different labels despite having the same contents. This causes problems in the evaluations. If the pairs have different labels, it is impossible to classify both of them correctly. If the pairs have the same label, a kNN classifier places more emphasis on this class for no reason. We report this issue in more detail in Appendix A.

The datasets released by the WMD paper have been used in many studies (Huang et al., 2016; Yurochkin et al., 2019; Le et al., 2019; Werner & Laber, 2020; Wang et al., 2020b; Wu et al., 2018; Skianis et al., 2020b; Mollaysa et al., 2017; Gupta et al., 2020; Skianis et al., 2020a) with the same protocol. We suspect that many readers and authors were unaware of the duplicate samples, which might have led to a misleading analysis. We release the indices of duplicate documents and a preprocessing script to remove duplicate samples for the following studies. We believe that sharing this fact within the community is considerably important for aiding in a solid evaluation and analysis.

In the following, we first use the same dataset as the WMD paper to highlight the essential differences with the original evaluation. We then evaluate using clean datasets to further corroborate the findings.

**Embeddings.** We use the same word embeddings as in the original paper (Kusner et al., 2015). Namely, we use the 300-dimentional word2vec embeddings trained on the Google News corpus. We found the second misleading point here.

**Misleading Point 2** The official code of WMD[2] normalized the embeddings by the L2 norm, although this was not explicitly stated in the paper.

The hint of word normalization is not in the main logic but only in the preprocessed binary file in the WMD's repository. We accidentally found this when we were debugging the code. Note that the Word Rotator's Distance (Yokoi et al., 2020) proposed to use the angles between word embeddings

---

[2] https://github.com/mkusner/wmd

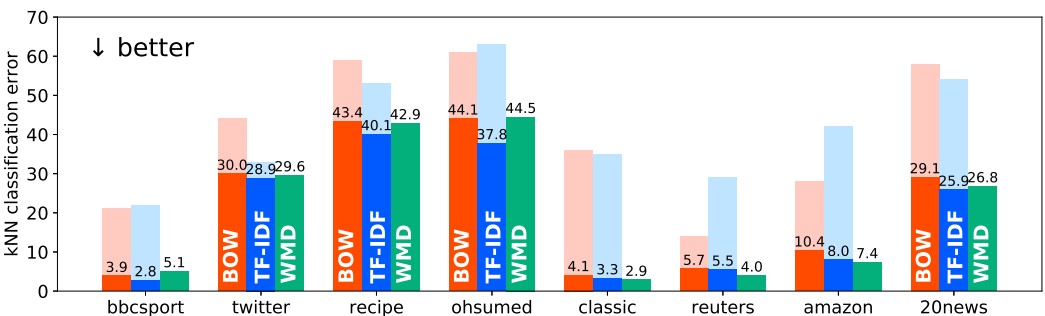

Figure 3: kNN classification errors in our re-evaluation. Lower is better. The shaded bars are the performance without normalization. WMD is comparable to classical baselines with normalization.

for the cost matrix of WMD, but in fact, the original WMD already used angles for the cost matrix. We suspect that most readers missed this point. In this paper, we follow the original evaluation and normalize the embeddings using the L2 norm.

**Preprocessing.** For WMD, we use the same preprocessing as in the original paper (Kusner et al., 2015), whereas for BOW and TF-IDF, we use a different preprocessing to clearly contrast the results. WMD discards certain words because the word embeddings do not contain all words. In the original study, the authors discard out-of-vocabulary words for WMD but maintain them for BOW and TF-IDF. In this paper, we discard out-of-vocabulary words for BOW and TF-IDF as well and use the same vocabulary for all of WMD, BOW, and TF-IDF. This setting is slightly advantageous for WMD. We show that WMD is comparable to BOW and TF-IDF even under this setting. We include the results with out-of-vocabulary words at the end of Section 4 for completeness.

**Evaluation Protocol.** WMD, BOW, and TF-IDF all have only one hyperparameter, i.e., size $k$ of the neighborhood in kNN classification. We split the training set into an 80/20 train/validation set uniformly and randomly and select the neighborhood size from $\{1, 2, \cdots, 19\}$ using the validation data. During the process of our re-evaluations, we found that the kNN classifiers were much less capable than we previously thought and that kNN classification may underestimate the performances of distance-based methods. Nevertheless, we adopt kNN evaluations in the main part to clearly contrast our results with the original evaluation. This is justified because we only use kNN classification for all methods. However, this fact can be a misleading point if other classifiers are used for other methods, as in (Wu et al., 2018; Skianis et al., 2020b; Mollaysa et al., 2017; Gupta et al., 2020). We report this issue in more detail in Appendix B.

**Environment.** We use a server cluster to compute WMD. Each node has two 2.4GHz Intel Xeon Gold 6148 CPUs. We use a Linux server with Intel Xeon E7-4830 v4 CPUs to evaluate the performances.

## 4    NORMALIZATION IS CRUCIAL

**Importance of Normalization.** In the original paper, raw BOW and TF-IDF vectors are used for the nearest neighbor classification. However, this is problematic because the length of these vectors varies based on the length of the documents. Even if two documents share the same topic, the BOW vectors are distant if their lengths differ. We need to normalize these vectors to effectively use them in the nearest neighbor classification.

**Misleading Point 3**  The original evaluation did not normalize the BOW and TF-IDF vectors.

To make direct comparisons to WMD, we employ L1 normalization (Eq. (1)) and the L1 distance for BOW and TF-IDF vectors, i.e., $d_{\text{BOW/L1/L1}}(\boldsymbol{x}, \boldsymbol{x}') = \|\boldsymbol{n}_{\text{L1}}(\boldsymbol{x}) - \boldsymbol{n}_{\text{L1}}(\boldsymbol{x}')\|_1$. With this normalization, BOW is a special case of WMD that does not use the underlying geometry. Specifically, let $\boldsymbol{C}^{\text{unif}}$ be the cost matrix of WMD when we use one-hot vectors as word embeddings, i.e., $\boldsymbol{C}_{ij}^{\text{unif}} = 0$ if $i = j$ and $\boldsymbol{C}_{ij}^{\text{unif}} = 2$ if $i \neq j$. Then,

**Proposition 1.**  $d_{\text{BOW/L1/L1}}(\boldsymbol{x}, \boldsymbol{x}') = \text{OT}(\boldsymbol{x}, \boldsymbol{x}', \boldsymbol{C}^{\text{unif}})$

Table 2: kNN classification errors. Lower is better. Here, (x/y) uses x as the normalization and y as the metric. The last column reports the average relative performance to the normalized BOW. These values correspond to Figure 4 in the original paper, but we use BOW (L1/L1) as the base performances, while the original paper used BOW (None/L2) as the base performances. Figure 4 in (Yurochkin et al., 2019) also reports the relative performances, but it uses BOW (L1/L2) as the base performances. The standard deviations are reported for five-fold datasets. The first three rows are the same as in Figure 3. For BOW and TF-IDF, a cell is highlighted with **bold** if the mean score is better than that of WMD. For WMD, a cell is highlighted with **bold** if the mean score is better than those of both BOW and TF-IDF. The fourth row reports the performances with WMD with TF-IDF weighting. The following rows report the performance with different normalization and metrics.

| | bbcsport | twitter | recipe | ohsumed | classic | reuters | amazon | 20news | rel. |
|---|---|---|---|---|---|---|---|---|---|
| BOW (L1/L1) | **3.9 ± 1.1** | 30.0 ± 1.1 | 43.4 ± 0.8 | **44.1** | 4.1 ± 0.5 | 5.7 | 10.4 ± 0.5 | 29.1 | 1.000 |
| TF-IDF (L1/L1) | **2.8 ± 1.1** | **28.9 ± 0.8** | **40.1 ± 0.7** | **37.8** | 3.3 ± 0.4 | 5.5 | 8.0 ± 0.3 | **25.9** | 0.861 |
| WMD | 5.1 ± 1.2 | 29.6 ± 1.5 | 42.9 ± 0.8 | 44.5 | **2.9 ± 0.4** | **4.0** | **7.4 ± 0.5** | 26.8 | 0.917 |
| WMD-TF-IDF | 3.3 ± 0.9 | 28.3 ± 2.3 | 39.9 ± 1.1 | 39.7 | 2.7 ± 0.3 | 4.0 | 6.6 ± 0.2 | 24.1 | 0.804 |
| BOW (None/L2)[Kusner et al. (2015)] | 19.4 ± 3.0 | 34.2 ± 0.6 | 60.0 ± 2.3 | 61.6 | 35.0 ± 0.9 | 11.8 | 28.2 ± 1.0 | 57.7 | 3.024 |
| BOW (None/L1) | 25.4 ± 1.5 | 32.7 ± 1.6 | 65.8 ± 2.5 | 69.3 | 52.1 ± 0.5 | 14.2 | 31.4 ± 1.2 | 73.9 | 3.931 |
| TF-IDF (None/L2)[Kusner et al. (2015)] | 24.5 ± 1.3 | 38.2 ± 4.6 | 65.0 ± 1.9 | 65.3 | 38.8 ± 1.0 | 28.0 | 41.2 ± 3.2 | 60.0 | 3.867 |
| TF-IDF (None/L1) | 30.6 ± 1.3 | 37.8 ± 4.8 | 70.3 ± 1.3 | 70.6 | 52.6 ± 0.2 | 29.1 | 41.5 ± 4.9 | 74.6 | 4.602 |
| BOW (L1/L2)[Yurochkin et al. (2019)] | 11.4 ± 3.6 | 37.0 ± 1.4 | 50.8 ± 1.1 | 56.7 | 17.3 ± 1.5 | 12.3 | 35.7 ± 1.3 | 46.5 | 2.253 |
| BOW (L2/L1) | 15.2 ± 1.5 | 33.3 ± 1.1 | 61.1 ± 1.1 | 65.7 | 51.1 ± 0.4 | 16.2 | 32.2 ± 1.3 | 77.6 | 3.622 |
| BOW (L2/L2)[Werner & Laber (2020)][Wrzalik & Krechel (2019)] | 5.5 ± 0.7 | 31.0 ± 0.8 | 46.1 ± 0.6 | 46.2 | 6.3 ± 0.7 | 8.8 | 13.1 ± 0.5 | 33.2 | 1.254 |
| TF-IDF (L1/L2) | 25.5 ± 11.2 | 35.7 ± 1.4 | 54.2 ± 2.7 | 61.4 | 22.6 ± 4.2 | 24.7 | 41.9 ± 2.0 | 45.6 | 3.226 |
| TF-IDF (L2/L1) | 27.5 ± 7.2 | 33.4 ± 1.7 | 64.9 ± 3.8 | 69.7 | 52.0 ± 0.2 | 19.5 | 40.8 ± 6.6 | 78.3 | 4.245 |
| TF-IDF (L2/L2)[Yurochkin et al. (2019)][Li et al. (2019)] | 4.0 ± 0.7 | 29.8 ± 1.5 | 43.7 ± 1.2 | 38.4 | 5.2 ± 0.3 | 10.5 | 11.1 ± 0.9 | 31.6 | 1.145 |

The proof is in Appendix E. This proposition shows that the difference in the performances between WMD and L1/L1 BOW indicates the benefit of the underlying geometry.

Figure 3 shows the classification errors with normalization. First, we can see that the errors of BOW and TF-IDF drastically decrease. Even BOW performs better than WMD in bbcsport and ohsumed. TF-IDF outperforms WMD in five out of eight datasets. Even in the other datasets where WMD outperforms the baselines, the improvements are far less significant than what was reported in the original evaluation. We can also observe that TF-IDF always performs better than BOW in contrast to the original evaluation. These results make more sense than what was reported in the WMD paper, where BOW outperformed TF-IDF.

**Comparison with other Normalization.** Although normalized BOW and TF-IDF have been employed in other studies (Yurochkin et al., 2019; Li et al., 2019; Werner & Laber, 2020; Wrzalik & Krechel, 2019), it was reported that normalized BOW is still far worse than WMD, which is incompatible with our observations above. We found out that this occurred because of the normalization methods and metrics used to compare the vectors. For example, Yurochkin et al. (2019) used L1 normalization and the L2 metric for BOW, i.e., $\|\boldsymbol{n}_{\mathrm{L1}}(\boldsymbol{x}) - \boldsymbol{n}_{\mathrm{L1}}(\boldsymbol{x}')\|_2$, and L2 normalization and the L2 metric for TF-IDF, i.e., $\|\boldsymbol{n}_{\mathrm{L2}}(\boldsymbol{x}) - \boldsymbol{n}_{\mathrm{L2}}(\boldsymbol{x}')\|_2$, where $\boldsymbol{n}_{\mathrm{L2}}(\boldsymbol{x}) = \boldsymbol{x}/\|\boldsymbol{x}\|_2$. Note that the L2/L2 scheme corresponds to the cosine similarity. We investigate the performance with different normalization and metrics using the same protocol as in the previous experiments. Table 2 shows the classification errors. First, it is easy to see in the fifth through eighth rows that the performances degrade without normalization. In addition, even with normalization, the performances are still poor if the normalization and metric use different norms. Although the L2/L2 scheme is better than these schemes, the L1/L1 scheme is the best for all datasets. This result is natural if we adopt the stance that a document is a distribution of words, and based on the analogy between the L1/L1 BOW and WMD, i.e., Proposition 1. To summarize, the normalization method significantly affects the performance. BOW L1/L1 corresponds to "WMD without OT" (Proposition 1), and BOW None/L1 corresponds "BOW L1/L1 without normalization". The performances of these methods are

$$3.931 \text{ (BOW None/L1)} \xrightarrow{\text{introducing normalization}} 1.000 \text{ (BOW L1/L1)} \xrightarrow{\text{introducing OT}} 0.917 \text{ (WMD)}$$

This means that WMD owes its performance improvements from the naive method (i.e., BOW None/L1) to normalization (by a factor of 3.9) rather than the OT formulation (by only a factor of 1.09). This "ablation study" indicates that L1 normalization is the most significant factor of improvement in WMD. We stress that the improvements brought about by L1 normalization are obtained almost for

Table 3: kNN classification errors with clean data. Lower is better. The same notations as in Table 2.

|  | bbcsport | twitter | recipe | ohsumed | classic | reuters | amazon | 20news | rel. |
|---|---|---|---|---|---|---|---|---|---|
| BOW (L1/L1) | **3.7 ± 1.0** | **30.6 ± 1.1** | **42.9 ± 0.6** | **39.7** | 4.2 ± 0.5 | 5.5 | 10.6 ± 0.6 | 29.2 | 1.000 |
| TF-IDF (L1/L1) | **2.3 ± 1.4** | **30.2 ± 0.7** | **40.0 ± 1.1** | **33.4** | 3.5 ± 0.2 | 5.9 | 8.0 ± 0.6 | **25.9** | 0.866 |
| WMD | 5.5 ± 1.2 | 30.6 ± 1.2 | 42.9 ± 0.9 | 40.6 | **3.4 ± 0.6** | **3.8** | **7.3 ± 0.4** | 26.9 | 0.952 |
| WMD-TF-IDF | 4.1 ± 1.5 | 28.8 ± 1.6 | 40.2 ± 0.9 | 35.7 | 2.8 ± 0.3 | 4.3 | 6.6 ± 0.3 | 24.2 | 0.848 |

Table 4: kNN classification errors with all words and the original datasets. Lower is better. The last column reports the average relative performance to the normalized BOW (the first row in Table 2).

|  | bbcsport | twitter | ohsumed | classic | reuters | amazon | 20news | rel. |
|---|---|---|---|---|---|---|---|---|
| BOW (L1/L1) | 3.0 ± 0.8 | 29.5 ± 0.7 | 46.0 | 3.9 ± 0.4 | 6.1 | 12.3 ± 0.5 | 32.1 | 1.015 |
| TF-IDF (L1/L1) | 2.8 ± 0.8 | 29.4 ± 0.9 | 38.7 | 3.1 ± 0.4 | 6.8 | 7.9 ± 0.3 | 24.8 | 0.877 |

Table 5: kNN classification errors with all words and clean data. Lower is better. The last column reports the average relative performance to the normalized BOW (the first row in Table 3).

|  | bbcsport | twitter | ohsumed | classic | reuters | amazon | 20news | rel. |
|---|---|---|---|---|---|---|---|---|
| BOW (L1/L1) | 2.7 ± 0.6 | 31.0 ± 2.0 | 41.0 | 4.1 ± 0.4 | 6.3 | 12.3 ± 0.3 | 32.1 | 1.022 |
| TF-IDF (L1/L1) | 1.8 ± 0.8 | 30.3 ± 1.5 | 34.9 | 3.4 ± 0.5 | 6.4 | 8.1 ± 0.2 | 24.8 | 0.849 |

free, whereas the improvements by WMD are at the price of substantial computational costs. We should also point out that many papers have not clarified the normalization scheme. Because the normalization method and metrics used to compare the vectors significantly affect the performance, we recommend clarifying them in the experimental setups.

To better illustrate the benefit of WMD, we also use TF-IDF weighting for the marginal measures of WMD. Specifically, let $x_{\text{tfidf}} \in \mathbb{R}^d$ denote a TF-IDF vector. We use $n_{\text{L1}}(x_{\text{tfidf}})$ instead of $n_{\text{L1}}(x)$ in the marginal constraints in Eq. (2). The TF-IDF weighting makes WMD as robust to noise as TF-IDF. The fourth row in Table 2 shows that WMD with TF-IDF weighting performs the best, with approximately a six percent improvement from the normalized TF-IDF, which corresponds to the WMD-TF-IDF with the uniform distance matrix $C^{\text{unif}}$. Investigating the (BOW, WMD) and (TF-IDF, WMD-TF-IDF) pairs illustrates the benefit of the optimal transport formulation. We can observe six to eight percent improvements in the optimal transport formulation from the naive baselines. This is in contrast to the *sixty* percent improvement claimed in the original study.

**Evaluation on Clean Data.** We evaluate the methods using clean datasets without duplicate samples. We adopt the same protocol as in the previous experiments except for the removal of duplicate documents. Table 3 reports the classification errors. Although the tendency is the same as that in the previous experiments, the differences between (BOW, WMD) and (TF-IDF, WMD-TF-IDF) become even narrower. Namely, an approximately five percent improvement in WMD and two percent improvement in WMD-TF-IDF are found. The results for other normalization and metrics are reported in Table 7 in the appendix.

**Evaluation with Out of Vocabulary Words.** We had hypothesized that the two to eight percent improvements of WMD were due to the unnecessarily discarded vocabularies of BOW, which made the evaluations slightly advantageous to WMD. We evaluate BOW and TF-IDF, including words not in the word2vec vocabulary. We use all but the recipe dataset because the raw texts of the recipe dataset have not been released by the authors. We include stopwords for the Twitter dataset and remove them for the other datasets following the original paper. Tables 4 and 5 report the classification errors. Against expectations, these results show that the use of all words does not improve the performance. We hypothesize that this is because the word2vec vocabulary implicitly helps the classification by removing noisy words.

**Summary.** We should emphasize that we have not concluded that the improvements brought by WMD are spurious. Based on our careful evaluations, we conclude that the two to eight percent improvements from the naive baselines reported in Tables 2 and 3 are genuine. These improvements are less sensational than those reported in the original paper, i.e., *sixty* percent improvement. However, we believe that our results indicate the true performance of WMD. These results indicate that if the speed is important, WMD may not be worth trying, whereas if the performance is crucial at any cost, WMD may be a worthwhile candidate over L1/L1 BOW.

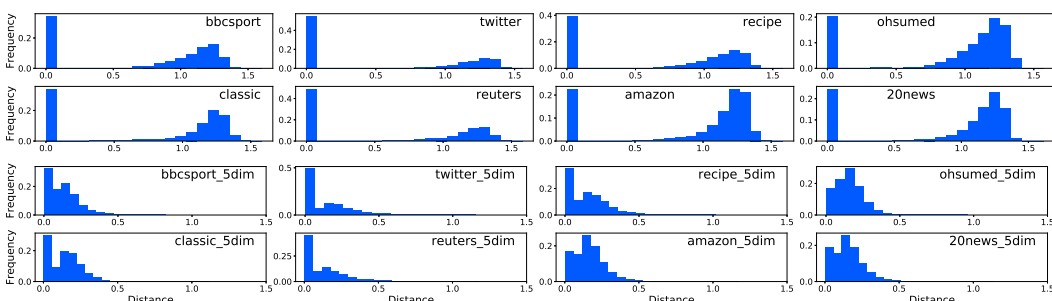

Figure 4: Histograms of distances between matched word embeddings. (Top) 300-dimensional embeddings. (Bottom) 5-dimensional embeddings.

## 5 WMD RESEMBLES BOW IN HIGH DIMENSIONAL SPACES

In this section, we experimentally show that not only the performance of WMD but also the distance values of WMD themselves resemble those of L1/L1 BOW.

**Travel Distance Distribution is Two-modal in High Dimensional Spaces.** We had assumed that the distribution of distances of matched word embeddings in WMD was a Gaussian-like one modal distribution. However, against expectations, we found that the distribution was actually two-modal in high dimensional spaces.

The top panels in Figure 4 show the histograms for matched words in nearest neighbor document pairs. The x-axis represents the distances of matched words, and the y-axis represents the frequency. All of the histograms have acute peaks at $x = 0$, where the source and target words exactly match. Other pairs of matched words are at $x \approx 1$ and are approximately equally distant. This occurs intuitively because most pairs of embeddings are almost orthogonal in high dimensional spaces.

To contrast the results with low dimensional cases, we project the 300-dimensional word2vec to a 5-dimensional space using principal component analysis and compute the same histograms using these low dimensional embeddings. Bottom panels in Figure 4 show the histograms. In contrast to the high dimensional cases, these histograms are almost one modal. These results indicate that the two modalities are a characteristic phenomenon in high dimensional spaces.

**WMD resembles L1/L1 BOW in High Dimensional Spaces.** The previous experiments show that the travel distance distribution is two modal (zero or around one) in high dimensional spaces. In an extreme case, if the distance is exactly zero or one, Proposition 1 shows that WMD coincides with L1/L1 BOW. If not exact, we experimentally show that the distance values of WMD themselves resemble those of L1/L1 BOW.

Figure 5 shows correlations between WMD and L1/L1 BOW distances and reports the Pearson's correlation coefficients $\rho$. WMD and L1/L1 BOW are surprisingly similar to each other. To contrast the results with low dimensional cases, we also report the scatter plots using 5-dimensional embeddings in the bottom of Figure 5. In contrast to the high dimensional cases, these scatter plots show that WMD in low dimensional spaces is not similar to L1/L1 BOW. These results indicate that the similarity to L1/L1 BOW is a characteristic phenomenon in high dimensional spaces.

For example, the distance between "Obama" and "President" is $1.174$, and the distance between "Obama" and "band" is $1.342$ in 300-dimensional word2vec. The distance between "speaks" and "greets" is $0.978$, and the distance between "speaks" and "gave" is $1.309$. Although "Obama" and "President" seem much more semantically similar than "Obama" and "band", the distances are not much different in high dimensional spaces. In other words, WMD does not identify "Obama" with "President" in high dimensional spaces. A simple calculation shows that

$$2\text{WMD}(\text{``Obama greets''}, \text{``band greets''}) = 1.342 + 0$$
$$< 2\text{WMD}(\text{``Obama greets''}, \text{``President speaks''}) = 1.174 + 0.978,$$
$$\text{BOW}(\text{``Obama greets''}, \text{``band greets''}) = 1$$
$$< \text{BOW}(\text{``Obama greets''}, \text{``President speaks''}) = 2.$$

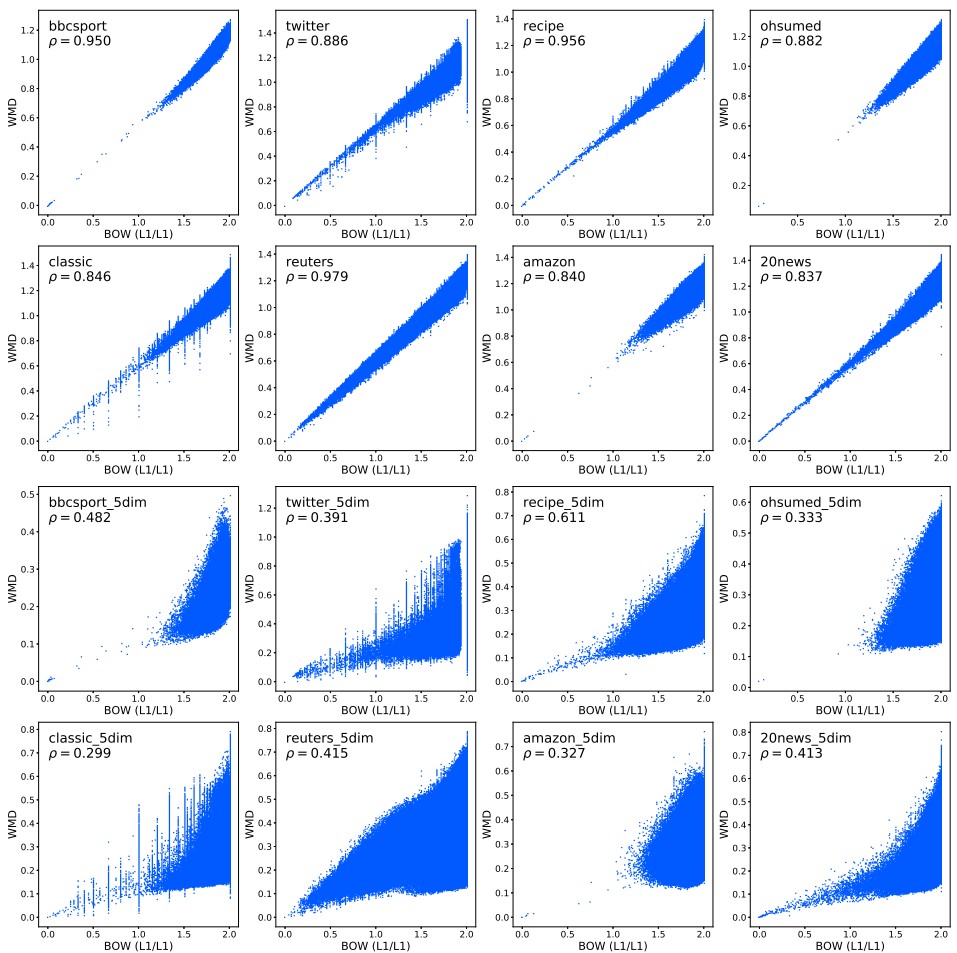

Figure 5: Scatter plot of WMD and L1/L1 BOW. (Top) 300-dimensional embeddings. WMD is mostly determined by L1/L1 BOW. (Bottom) 5-dimensional embeddings.

This "almost equally distant" property may not be expected in the two-dimensional illustration (Figure 1). While such an illustration is helpful for understanding the mechanism of WMD, it does not reflect the high dimensional nature of word embeddings. The characteristics of high dimensionality should also be kept in mind to understand the behavior of WMD more precisely.

It should be noted that 5-dimensional WMD performs worse than 300-dimensional WMD because 5-dimensional WMD loses much information on the word geometry. On the one hand, by increasing the number of dimensions, the word embeddings become more discriminative, and the performance of WMD increases. On the other hand, by increasing the number of dimensions, the word embeddings become orthogonal, and WMD approaches L1/L1 BOW, i.e., a special case of WMD that completely discriminates each word from the others.

# 6 CONCLUSION

In this paper, we pointed out that the major improvement in WMD against classical baselines is not from the inherent feature of WMD but mainly from the normalization. We re-evaluated the performance of WMD and classical baselines and found that classical baselines are competitive with WMD if we normalize the vectors. We also found that WMD resembled BOW much more than we had thought in two-dimensional illustrations owing to the high dimensionality of the word embeddings. In the process of our re-evaluation, we found several confusing aspects in the original evaluations of the original paper on WMD. We pointed them out in this paper to help the following researchers conduct solid evaluations.

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

Table 6: Weighted kNN classification errors. Lower is better. The last column reports the average relative performance to the normalized BOW with the standard kNN (the first row in Table 2).

| | bbcsport | twitter | recipe | ohsumed | classic | reuters | amazon | 20news | rel. |
|---|---|---|---|---|---|---|---|---|---|
| BOW (L1/L1/wkNN) | $3.3 \pm 0.6$ | $26.4 \pm 1.6$ | $41.1 \pm 0.5$ | 41.3 | $3.6 \pm 0.4$ | 4.7 | $10.3 \pm 0.3$ | 23.3 | 0.888 |
| TF-IDF (L1/L1/wkNN) | $2.0 \pm 0.7$ | $26.6 \pm 1.5$ | $38.3 \pm 1.1$ | 36.1 | $2.8 \pm 0.4$ | 5.8 | $8.2 \pm 0.5$ | 20.5 | 0.787 |
| WMD (wkNN) | $4.4 \pm 1.2$ | $26.2 \pm 2.2$ | $40.5 \pm 1.0$ | 41.1 | $2.7 \pm 0.4$ | 3.6 | $7.1 \pm 0.5$ | 21.7 | 0.823 |
| WMD-TF-IDF (wkNN) | $3.2 \pm 1.0$ | $25.6 \pm 1.5$ | $37.7 \pm 0.9$ | 37.0 | $2.4 \pm 0.4$ | 4.0 | $6.3 \pm 0.4$ | 19.6 | 0.743 |

## A    DUPLICATE DOCUMENTS

We point out that there are many duplicate samples in the datasets used in the original study. The last two rows in Table 2 report the numbers of duplicate samples. Specifically, "duplicate pair" reports the number of pairs $(s, t)$ such that $s$ and $t$ are the same, and "duplicate samples" reports the number of samples $s$ such that $s$ has at least one samples $t$ with the same content. Some of these pairs cross the training and test splits, and some of them have different labels despite having the same contents. This causes problems in the evaluations. If the pairs have different labels, it is impossible to classify both of them correctly. If the pairs have the same label, a kNN classifier places more emphasis on this class for no reason. We found that duplication was caused by different reasons in different datasets. For example, the ohsumed dataset was originally a multi-labeled dataset, and the WMD paper duplicate samples for each label. We found that the original source dataset (Greene & Cunningham, 2006) of bbcsport[3] already contained duplicate samples, e.g., `athletics/12.txt` and `athletics/20.txt`. We hypothesis this was originated from the data collection process.

Although such duplication is not a problem if we intend to measure the performances in noisy environments, the analysis could be misleading if we assume a clean environment. The datasets released by the WMD paper have been used in many studies (Huang et al., 2016; Yurochkin et al., 2019; Le et al., 2019; Werner & Laber, 2020; Wang et al., 2020b; Wu et al., 2018; Skianis et al., 2020b; Mollaysa et al., 2017; Gupta et al., 2020; Skianis et al., 2020a) with the same protocol. We suspect that many readers and authors were unaware of this fact, which might have led to a misleading analysis. We release the indices of duplicate documents and a preprocessing script to remove duplicate samples for the following studies. We believe that sharing this fact within the community is considerably important for aiding in a solid evaluation and analysis.

## B    NEAREST NEIGHBOR CLASSIFICATION IS FAR FROM OPTIMAL

WMD is often used as a weak baseline in document classification. In this section, we point out that the poor performances observed in previous studies are not necessarily due to the problem we found in the main part but rather due to the choices of classifiers. We found that many existing studies, including the original paper on WMD, used kNN classification for distance-based methods (Kusner et al., 2015; Huang et al., 2016; Wang et al., 2020b). This is reasonable owing to its simplicity regardless of suboptimal performance. However, we found that kNN classification was much farther from optimal than we had thought and caused some improper observations. The most prominent issue appears in comparison with non-kNN classifiers. For example, WMD is used with a kNN classifier as a baseline, and other classifiers are adopted for the proposed methods in (Wu et al., 2018; Skianis et al., 2020b; Mollaysa et al., 2017; Li et al., 2019; Gupta et al., 2020; Nikolentzos et al., 2020; 2017). The improvement observed by such evaluations may not be due to the superiority of the proposed methods against WMD, but due to the superiority of the classifier against the kNN classifier. To mitigate this problem, we found that a weighted kNN classifier (wkNN) with exponential weighting was a good choice for distance-based methods. Specifically, wkNN first gathers $k$ nearest neighbor samples and computes the distances $\{d_1, \cdots, d_k\}$ to these samples. It then defines the weight for sample $i$ as $\exp(-d_i/\gamma)$ and takes the weighted majority vote, where $\gamma$ is a hyperparameter. We can think of wkNN as a continuous variant of kNN, which uses a step function as the weight function. As the crux of wkNN, the time and space complexities are the same as in kNN, and it involves no training procedures. Thus, we can easily replace a kNN system with a wkNN system.

---

[3] http://mlg.ucd.ie/datasets/bbc.html

Table 7: kNN classification errors with clean data. Lower is better. The same notations as in Table 2.

| | bbcsport | twitter | recipe | ohsumed | classic | reuters | amazon | 20news | rel. |
|---|---|---|---|---|---|---|---|---|---|
| BOW (L1/L1) | $\mathbf{3.7 \pm 1.0}$ | $\mathbf{30.6 \pm 1.1}$ | $\mathbf{42.9 \pm 0.6}$ | 39.7 | $4.2 \pm 0.5$ | 5.5 | $10.6 \pm 0.6$ | 29.2 | 1.000 |
| TF-IDF (L1/L1) | $\mathbf{2.3 \pm 1.4}$ | $\mathbf{30.2 \pm 0.7}$ | $\mathbf{40.0 \pm 1.1}$ | 33.4 | $3.5 \pm 0.2$ | 5.9 | $8.0 \pm 0.6$ | $\mathbf{25.9}$ | 0.866 |
| WMD | $5.5 \pm 1.2$ | $30.6 \pm 1.2$ | $42.9 \pm 0.9$ | 40.6 | $\mathbf{3.4 \pm 0.6}$ | $\mathbf{3.8}$ | $\mathbf{7.3 \pm 0.4}$ | 26.9 | 0.952 |
| WMD-TF-IDF | $4.1 \pm 1.5$ | $28.8 \pm 1.6$ | $40.2 \pm 0.9$ | 35.7 | $2.8 \pm 0.3$ | 4.3 | $6.6 \pm 0.3$ | 24.2 | 0.848 |
| BOW (None/L2) | $22.8 \pm 1.6$ | $34.2 \pm 0.6$ | $59.1 \pm 0.9$ | 60.7 | $36.9 \pm 1.1$ | 11.7 | $28.9 \pm 1.1$ | 58.0 | 3.227 |
| BOW (None/L1) | $25.3 \pm 2.1$ | $33.9 \pm 0.8$ | $64.1 \pm 0.8$ | 67.4 | $55.0 \pm 0.5$ | 14.0 | $32.4 \pm 1.3$ | 73.5 | 4.044 |
| TF-IDF (None/L2) | $25.8 \pm 1.1$ | $33.5 \pm 0.6$ | $65.6 \pm 1.0$ | 62.8 | $41.1 \pm 1.1$ | 28.3 | $43.4 \pm 5.4$ | 59.8 | 4.032 |
| TF-IDF (None/L1) | $32.7 \pm 1.2$ | $33.6 \pm 0.6$ | $70.6 \pm 1.8$ | 69.5 | $55.6 \pm 0.2$ | 29.1 | $42.5 \pm 4.8$ | 74.7 | 4.804 |
| BOW (L1/L2) | $11.8 \pm 0.7$ | $40.2 \pm 2.0$ | $51.4 \pm 1.4$ | 55.8 | $17.5 \pm 1.7$ | 12.9 | $36.8 \pm 1.4$ | 46.7 | 2.336 |
| BOW (L2/L1) | $15.0 \pm 1.6$ | $34.3 \pm 0.9$ | $59.9 \pm 0.7$ | 64.3 | $54.0 \pm 0.4$ | 16.2 | $32.7 \pm 2.1$ | 77.1 | 3.715 |
| BOW (L2/L2) | $5.3 \pm 0.9$ | $32.6 \pm 0.7$ | $45.4 \pm 1.1$ | 43.2 | $6.7 \pm 0.7$ | 8.1 | $13.3 \pm 0.5$ | 33.2 | 1.263 |
| TF-IDF (L1/L2) | $20.0 \pm 7.4$ | $39.3 \pm 1.4$ | $54.1 \pm 2.4$ | 55.2 | $24.8 \pm 3.0$ | 24.5 | $43.2 \pm 2.0$ | 45.9 | 3.168 |
| TF-IDF (L2/L1) | $28.7 \pm 5.8$ | $33.5 \pm 1.0$ | $65.7 \pm 2.8$ | 65.7 | $55.0 \pm 0.2$ | 19.9 | $39.0 \pm 5.4$ | 78.6 | 4.390 |
| TF-IDF (L2/L2) | $3.1 \pm 1.3$ | $31.5 \pm 1.5$ | $43.6 \pm 0.8$ | 33.2 | $5.2 \pm 0.4$ | 10.6 | $11.1 \pm 0.6$ | 31.6 | 1.127 |

Table 8: kNN classification errors with 5 dimensional embeddings. Lower is better. The first two rows are the same as Table 2 for reference.

| | bbcsport | twitter | recipe | ohsumed | classic | reuters | amazon | 20news | rel. |
|---|---|---|---|---|---|---|---|---|---|
| BOW (L1/L1) | $3.9 \pm 1.1$ | $30.0 \pm 1.1$ | $43.4 \pm 0.8$ | 44.1 | $4.1 \pm 0.5$ | 5.7 | $10.4 \pm 0.5$ | 29.1 | 1.000 |
| WMD | $5.1 \pm 1.2$ | $29.6 \pm 1.5$ | $42.9 \pm 0.8$ | 44.5 | $2.9 \pm 0.4$ | 4.0 | $7.4 \pm 0.5$ | 26.8 | 0.917 |
| WMD (5 dim) | $18.5 \pm 0.8$ | $32.0 \pm 1.0$ | $51.8 \pm 1.0$ | 60.2 | $6.2 \pm 0.3$ | 6.9 | $15.9 \pm 0.8$ | 45.4 | 1.773 |

We conduct experiments to show the suboptimality of kNN. As the drawback of wkNN, it has two hyperparameters $k$ and $\gamma$. We find that the $k$ of wkNN plays a similar role as the maximum $k$ of kNN in the hyperparameter tuning. Thus, we fix $k$ of wkNN to 19 and tune only $\gamma$ in the hyperparameter tuning. Recall that the hyperparameter candidates were $k \in \{1, \cdots, 19\}$ for kNN in the previous experiments and original paper. We select $\gamma$ from $\Gamma = \{0.005, 0.010, \cdots, 0.095, 0.1\}$ ($|\Gamma| = 20$) in the hyperparameter tuning[4]. The remaining settings are the same as in the previous experiments. Table 6 shows the classification errors. We can see that wkNN performs much better than kNN. The benefit of wkNN (the first row in Table 6 versus the first row in Table 2) is more significant than the benefit of WMD (the third versus the first row in Table 2). This indicates that if a proposed method uses non-kNN classifiers, ten percent improvements from the kNN baselines may be due to classifiers, not to the proposed method. This also indicates that there is significant room for improvement in the classifiers before undergoing a design of better similarity measures.

It might be acceptable if all baselines and the proposed method use kNN because the relative performances in Tables 2 and 6 do not change significantly. However, it would be better to use more capable classifiers because the results in a paper will be cited as baseline records in following papers, in which other classifiers may be employed. In fact, such comparisons have been carried out in (Wu et al., 2018; Skianis et al., 2020b; Mollaysa et al., 2017; Gupta et al., 2020). Besides, practitioners may underestimate the performance of distance-based methods based on the reported results. We find wkNN is a better choice than kNN because it is as fast as kNN yet much more effective.

## C    COMPARISON WITH OTHER NORMALIZATION WITH CLEAN DATA

Table 7 shows the kNN classification errors with clean data. The same tendency as Table 2 can be observed.

## D    PERFORMANCE OF LOW DIMENSIONAL EMBEDDINGS

Table 8 shows the classification errors of WMD with the 5-dimensional embeddings used in Section 5. It shows that WMD with the low dimensional embeddings performs worse than with the original 300-dimensional embeddings, although WMD performs differently from BOW with the low dimensional embeddings (Figure 5). In general, an exact match of words indicates a strong similarity. It is

---

[4]The magnitude of this range is determined by the empirical distances between samples. This arbitrary choice is a slight drawback of wkNN. We will investigate how to choose it automatically in future work.

reasonable that counting common words can perform better than relying too much on the underlying geometry.

## E    PROOF OF PROPOSITION 1

We prove the analogy between L1/L1 BOW and OT. Recall that
$$d_{\text{BOW/L1/L1}}(\boldsymbol{x}, \boldsymbol{x}') = \|\boldsymbol{n}_{\text{L1}}(\boldsymbol{x}) - \boldsymbol{n}_{\text{L1}}(\boldsymbol{x}')\|_1,$$
and
$$\boldsymbol{C}_{ij}^{\text{unif}} = \begin{cases} 0 & (i = j) \\ 2 & (i \neq j). \end{cases}$$

*Proof of Proposition 1.* Without loss of generality, we assume that the marginals $\boldsymbol{x}$ and $\boldsymbol{x}'$ are already L1 normalized, and $\sum_i \boldsymbol{x}_i = 1$ and $\sum_i \boldsymbol{x}'_i = 1$ hold. Let $\boldsymbol{P}^* \in \mathbb{R}^{m \times m}$ be the optimal solution of $\text{OT}(\boldsymbol{x}, \boldsymbol{x}', \boldsymbol{C}^{\text{unif}})$. From the marginal constraints, $\boldsymbol{P}_{ii}^* \leq \min(\boldsymbol{x}_i, \boldsymbol{x}'_i)$ holds. If $\boldsymbol{P}_{ii}^* < \min(\boldsymbol{x}_i, \boldsymbol{x}'_i)$, there exists $j \neq i$ and $k \neq i$ such that $\boldsymbol{P}_{ij}^* > 0$ and $\boldsymbol{P}_{ki}^* > 0$ from the marginal constraints. Then, let $\boldsymbol{Q} \in \mathbb{R}^{m \times m}$ be
$$\boldsymbol{Q}_{st} = \begin{cases} \boldsymbol{P}_{ii}^* + \min(\boldsymbol{P}_{ij}^*, \boldsymbol{P}_{ki}^*) & (s = i \wedge t = i) \\ \boldsymbol{P}_{ij}^* - \min(\boldsymbol{P}_{ij}^*, \boldsymbol{P}_{ki}^*) & (s = i \wedge t = j) \\ \boldsymbol{P}_{ki}^* - \min(\boldsymbol{P}_{ij}^*, \boldsymbol{P}_{ki}^*) & (s = k \wedge t = i) \\ \boldsymbol{P}_{kj}^* + \min(\boldsymbol{P}_{ij}^*, \boldsymbol{P}_{ki}^*) & (s = k \wedge t = j) \\ \boldsymbol{P}_{st}^* & (\text{otherwise}). \end{cases}$$
Then, $\boldsymbol{Q}$ satisfies the constraints of OT and
$$\begin{aligned} &\left(\sum_{st} \boldsymbol{C}_{st}^{\text{unif}} \boldsymbol{P}_{st}^*\right) - \left(\sum_{st} \boldsymbol{C}_{st}^{\text{unif}} \boldsymbol{Q}_{st}\right) \\ &= \boldsymbol{C}_{ii}^{\text{unif}}(\boldsymbol{P}_{ii}^* - (\boldsymbol{P}_{ii}^* + \min(\boldsymbol{P}_{ij}^*, \boldsymbol{P}_{ki}^*))) + \boldsymbol{C}_{ij}^{\text{unif}}(\boldsymbol{P}_{ij}^* - (\boldsymbol{P}_{ij}^* - \min(\boldsymbol{P}_{ij}^*, \boldsymbol{P}_{ki}^*))) \\ &\quad + \boldsymbol{C}_{ki}^{\text{unif}}(\boldsymbol{P}_{ki}^* - (\boldsymbol{P}_{ki}^* - \min(\boldsymbol{P}_{ij}^*, \boldsymbol{P}_{ki}^*))) + \boldsymbol{C}_{kj}^{\text{unif}}(\boldsymbol{P}_{kj}^* - \boldsymbol{P}_{kj}^* + \min(\boldsymbol{P}_{ij}^*, \boldsymbol{P}_{ki}^*))) \\ &= (4 - \boldsymbol{C}_{kj}^{\text{unif}}) \min(\boldsymbol{P}_{ij}^*, \boldsymbol{P}_{ki}^*) \\ &> 0. \end{aligned}$$
The first equation holds from the definition of $\boldsymbol{Q}$, and the second equation holds because $\boldsymbol{C}_{ii}^{\text{unif}} = 0$ and $\boldsymbol{C}_{ij}^{\text{unif}} = \boldsymbol{C}_{jk}^{\text{unif}} = 2$, and the last inequality holds because $\boldsymbol{C}_{kl}^{\text{unif}} \leq 2$ and $\min(\boldsymbol{P}_{ij}^*, \boldsymbol{P}_{ki}^*) > 0$. This contradicts with the optimality of $\boldsymbol{P}^*$. Therefore,
$$\boldsymbol{P}_{ii}^* = \min(\boldsymbol{x}_i, \boldsymbol{x}'_i)$$
holds. Therefore,
$$\begin{aligned} &\sum_{st} \boldsymbol{C}_{st}^{\text{unif}} \boldsymbol{P}_{st}^* \\ &= \sum_{st} 2(\mathbb{1}\mathbb{1}^\top - \text{diag}(\mathbb{1}))_{st} \boldsymbol{P}_{st}^* \\ &= 2 \sum_{st} \boldsymbol{P}_{st}^* - 2 \sum_s \boldsymbol{P}_{ss}^* \\ &= 2 - 2 \sum_s \min(\boldsymbol{x}_s, \boldsymbol{x}'_s) \\ &= \sum_s \boldsymbol{x}_s + \sum_s \boldsymbol{x}'_s - \sum_s \min(\boldsymbol{x}_s, \boldsymbol{x}'_s) - \sum_s \min(\boldsymbol{x}_s, \boldsymbol{x}'_s) \\ &= \sum_s (\boldsymbol{x}_s - \min(\boldsymbol{x}_s, \boldsymbol{x}'_s)) + (\boldsymbol{x}'_s - \min(\boldsymbol{x}_s, \boldsymbol{x}'_s)) \\ &= \sum_s |\boldsymbol{x}_s - \boldsymbol{x}'_s| \\ &= \|\boldsymbol{x} - \boldsymbol{x}'\|_1. \end{aligned}$$

$\square$

# F  ADDITIONAL RELATED WORK

We review additional related work, which we could not place in the main text because of the page limit. First, it has already been known that OT behaves pathologically in high dimensional spaces, mainly owing to the sample complexity. Dudley (1969) showed that the sample complexity of OT grows exponentially with respect to the number of dimensions. Weed et al. (2019) alleviated the bound using other senses of dimensionality, but the dependency is still exponential. Genevay et al. (2019) showed that the sample complexity of the Sinkhorn Divergences (Genevay et al., 2018; Cuturi, 2013) matches that of MMD. The crucial difference between our work and theirs is that we shed light on more practical sides, whereas their interests were on the theoretical side. We believe that our analysis and explanation are insightful, especially for practitioners.

Secondly, high-dimensional problems have also been studied from computational aspects. Rabin et al. (2011) and Kolouri et al. (2016) proposed to project embeddings to random one-dimensional spaces to speedup computation because OT in a one-dimensional space can be solved efficiently (Santambrogio, 2015). Paty & Cuturi (2019) and others (Kolouri et al., 2019; Dhouib et al., 2020; Petrovich et al., 2020; Lin et al., 2020; Huang et al., 2021) proposed robust variants of OT with low dimensional projections. Specifically, their methods adopt the worst distance with respect to candidate projections. Although we tried in early experiments several projection methods, including the principal component analysis as in Section 5 and Appendix D, tree slicing (Le et al., 2019), and rank-based cost matrices instead of actual distance cost matrices (Sato et al., 2020a), we did not find performance improvements. However, exploring these approaches can be one of the promising future directions with the curse of dimensionality in mind.

