# OpenReview forum: "Re-evaluating Word Mover's Distance"
_ICLR.cc/2022/Conference — ICLR 2022 Submitted_

### Official Review · Reviewer_sFeb · 2021-10-26

**Correctness:** 4
**Technical Novelty And Significance:** 1
**Empirical Novelty And Significance:** 3
**Recommendation:** 3
**Confidence:** 4

**Main Review:**

### Strength
The paper presents experimental results that the WMD distance doesn't improve performance by a factor of 100% but rather by a factor of 10%. This corrects some misconceptions that may have arisen in a particular sub-field of NLP.

### Weakness
The paper will have a low impact, and is too focused on refuting the results of a single paper. I believe that this paper may be well suited to conferences devoted to the particular application area of document classification or NLP in general, but as it is the paper's methodological contribution, or technical contribution is too little.

The paper presents a sub-section on the "re-evaluation of existing methods" in section 2 to answer some of this criticism, but all the papers in that section were much more wide-ranging in their focus-area. E.g. the (Dacrema et al.) paper was not focused on the results of a single paper but on multiple papers, and the paper by (Arora et al. 2017) presented a novel method. The paper by (Shen et al. 2018) was accepted at ACL which is an NLP conference more focused on NLP tasks.


**Summary Of The Paper:**

This paper shows that when standard baseline methods are carefully evaluated, such as using L1 normalization of the tf-idf/bow vector, then the performance difference between the baselines and the WMD distance based methods shrinks considerably.

The authors also claim that the findings in the paper are more general in the sense, that the distance between the embeddings of two words behaves more and more like a delta function, which is bimodal : close to one when the words are distinct, and close to 0 when the words are the same.


**Summary Of The Review:**

The paper corrects some errors made in an earlier paper by (Kusner et al. 2015) and presents a careful evaluation of baselines for document classification applications. However, I believe that the paper is more suited for an NLP conference than ICLR.

---

> ### Author Response · Authors · 2021-11-15
> **Official Author Response**
>
> Thank you for the comments. We try to address your concerns in the following.
>
> > too focused on refuting the results of a single paper
>
> We claim that our paper has a broad impact beyond a single paper.
>
> As for the misleading point 1, the datasets and the evaluation protocol are used in many papers, including Huan et al. NeurIPS 2016, Yurochkin et al. NeurIPS 2019, Le et al. NeurIPS 2019, Takezawa et al. ICML 2021, Wu et al. EMNLP 2018, Mollaysa et al. ICML 2017, Gupta et al. AAAI 2020, Skianis et al. AISTATS 2020. Clarifying the pitfalls affects many researches other than the original WMD paper.
>
> The misleading point 2 indeed confused the authors and readers of the word rotator's distance (Yokoi et al. EMNLP 2020).
>
> As for the misleading point 3, the non-effective normalization methods are used in many following papers, including Yurochkin et al. NeurIPS 2019, Li et al. WWW 2019, Werner et al. 2020, Wrzalik et al. 2019. The importance of reporting the issue is beyond a single paper here as well.
>
> These relevant papers indicate that our paper has a broad impact beyond a single paper.
>
> > as it is the paper's methodological contribution, or technical contribution is too little.
>
> Investigating the pitfalls of the evaluation protocol may not look sensational as, say, proposing a BERT-style model, but we believe it is important and necessary work for the community.
>
> > I believe that the paper is more suited for an NLP conference than ICLR.
>
> We claim that our paper is suitable for machine learning conferences as well. i) The original paper of WMD was presented in ICML. Discussing a paper presented in a machine learning conference in a machine learning conference is reasonable. ii) As we show above, many relevant papers have been published in machine learning conferences. iii) Not a few members in the ICLR community (at least Reviewer M3dN, Reviewer 5Jna, and we) are interested in the re-evaluation of WMD.

---

### Official Review · Reviewer_id2C · 2021-10-30

**Correctness:** 2
**Technical Novelty And Significance:** 1
**Empirical Novelty And Significance:** 1
**Recommendation:** 3
**Confidence:** 4

**Main Review:**

This paper presents a thorough re-evaluation of the word mover's distance paper by Kusner et al. (2015), focusing on a number of points that are considered misleading. The paper has a point about the fact that the normalization of the word vectors used in WMD by Kusner et al. was not mentioned in the paper, even though it was possible to find out in the code released by the authors (misleading point 2). I think that while welcome as a clarification, I think calling this omission misleading is a bit of a stretch, especially given that this was possible to find in the code released by the authors. The paper then criticises Kusner et al. for not using such a nornalization for the document vectors obtained by BOW and TFIDF, which they find improves their results (misleading point 3). However, it should be noted that the original paper included other methods for comparison, which performed rather competitive to WMD, so while I agree with this paper that Kusner et al. didn't do justice to BoW and TFIDF, they did show that other methods were competitive to WMD but not as good. This paper also admits this too. Furthermore, in WMD the normalization is at the word embedding level, while the L1/L2 normalization on BoW/TFIDF is at the document level, thus it is not a direct correspondence. Thus I don't think there is much to see here, unless the paper would like to be more of a case of how to make the most of token-matching distance metrics.

The first misleading point is about the datasets containing duplicates. But not sure this has to do that much with the method of Kusner et al. per se, especially given that the datasets are well-known and used frequently. Perhaps they shouldn't be, and a paper could be written about this. As for the misleading point 4, I have to disagree with this paper. The example from figure 1 in Kusner et al illustrates well how word embeddings could help us do something more useful than 0-1 distances in token matching metrics, which would be the case of their example. The arguments in  section 5 on the modes (not modalities) are not particularly tight. If anything, the BoW representation of a word or document is much more high dimensional than that of a (300 dim) word embedding, assuming that one has thousands of words typically. The point that Kusner et al. made was the word embedding distances are more informative than word matching, and this is confirmed in the plots of figure 4: different word pairs get different distances.

On the whole I believe that much of the criticism of this paper is not justified. The main claims of the Kusner et al. paper still hold, and the experiments in this paper confirm them in my mind. I would suggest that perhaps a better paper would be assessing the evaluation practices in distance metrics more broadly, using multiple state-of-the-art methods. While WMD influenced the way people are thinking, using it for this purpose with old embeddings on text classification which is nowadays dominated by fine-tuning pre-trained language models is not particularly informative.

**Summary Of The Paper:**

This paper presents a re-evaluation of a well known distance metric for documents with word embeddings. The authors identify some missing information from the original paper and they provide some extra analyses.

**Summary Of The Review:**

On the whole, while I appreciate the work to bring to the community's attention some extra information and analyses of the Word Mover's Distance paper, the paper is not very informative. I think a much more interesting paper would be to survey the evaluation of multiple distance metric learning approaches and discuss common pitfalls more broadly, rather than pick on one paper in the context of a task that it wouldn't be considered a standard method for (fine-tuning BERT style pre-trained models is the go-to method for text classification and is much better than word embeddings from 2015 with kNN).

---

> ### Author Response · Authors · 2021-11-15
> **Official Author Response**
>
> Thank you for the detailed comments. We try to address your concerns in the following.
>
> > I think calling this omission misleading is a bit of a stretch
>
> The word rotator's distance (Yokoi et al. EMNLP 2020) proposed to normalize the word embeddings, although it was already done in the original WMD paper. They did not notice that the word embeddings were normalized in the original WMD paper. This indicates that the omission indeed misleads the following research.
>
> > Furthermore, in WMD the normalization is at the word embedding level, while the L1/L2 normalization on BoW/TFIDF is at the document level, thus it is not a direct correspondence.
>
> WMD does both word-level and document-level normalization. The document level normalization in WMD corresponds to Eq. (1), i.e., WMD's regarding a document as a probability distribution. Proposition 1 shows the correspondence.
>
> > As for the misleading point 4, I have to disagree with this paper.
>
> There are two aspects here, (1) the word embeddings provide finer information than BoW, and (2) WMD behaves quite differently in high dimensional spaces from WMD in low dimensional spaces, as shown in Figures 4 and 5. We agree that Figure 1 elaborately illustrates the former point. What we are arguing in misleading point 4 is that Figure 1 overlooks the latter point. Thinking about WMD only with the two-dimensional illustration would overlook the characteristics of high dimensional embeddings. Note that we are not claiming Figure 1 should be omitted. Showing both Figure 1 (for the former point) and Figures 4 and 5 (for the latter point) would effectively describe the actual behavior of WMD.

---

> > ### Comment · Reviewer_id2C · 2021-11-19
> > **Response to author response**
> >
> > - on misleading point 1: I think we agree that the problem is the dataset; my objection is to why attribute this problem to Kusner et al and not to the paper proposing the dataset in the first place. I think it would be a fairer framing.
> >
> > - on misleading point 2: I don't object that it would have been better to have mentioned the normalization in the paper, but given that it is in the code, I think it is unfair to call it misleading. However, it is misleading on the part of this paper not to mention that Kusner et al compared to many other methods that perform much better than BoW. To offer an example of a paper that does in my opinion a good job in comparing neural and non-neural methods (in the case of word embeddings) and how lessons can be transferred, see this paper: https://levyomer.files.wordpress.com/2015/03/improving-distributional-similarity-tacl-2015.pdf
> >
> > - on misleading point 4: The figures 4 and 5 presented in the paper are dependent on the choice of word embeddings. Nothing wrong with this, but I don't think they are relevant to figure 1 from Kusner et al. which is a visual presentation of the intuition behind WMD. Again calling the figure 1 misleading is not appropriate in my opinion.

---

> > > ### Author Response · Authors · 2021-11-23
> > > **Official Author Response**
> > >
> > > Thank you for the response. We try to address the remaining concerns.
> > >
> > > > my objection is to why attribute this problem to Kusner et al and not to the paper proposing the dataset in the first place.
> > >
> > > As we stated in Appendix A, the duplicated samples were caused by several reasons. On the one hand, the original bbcsports data already contain several duplicated samples. On the other hand, although the ohsumed dataset originally contains no duplicated samples, the preprocessing of Kusner et al. introduced many (i.e., 3116) duplicated samples. The same preprocessing as Kusner et al. has been used by many following researchers (e.g., Huan et al. NeurIPS 2016, Yurochkin et al. NeurIPS 2019, Le et al. NeurIPS 2019, Takezawa et al. ICML 2021, Wu et al. EMNLP 2018, Mollaysa et al. ICML 2017, Gupta et al. AAAI 2020, Skianis et al. AISTATS 2020).
> > >
> > >
> > > > given that it is in the code, I think it is unfair to call it misleading
> > >
> > > We assume that Reviewer id2C has not checked where it is in the code. The hint of word normalization is not in the main logic but only in the preprocessed binary file in the WMD's GitHub repository. Although we accidentally found this when we were debugging the code, it is really difficult to find it without knowing it beforehand. Word rotator's distance indeed missed it, which indicates the difficulty. We've revised it and added the explanations.
> > >
> > >
> > > > it is misleading on the part of this paper not to mention that Kusner et al compared to many other methods that perform much better than BoW.
> > >
> > > We focused on BoW in this paper owing to the following reasons: (i) The performance of BoW is crucial because the performance score (Figure 4 in Kusner et al.) is defined based on the performance of BoW. The practice of using BoW as the base performance has been inherited by (Yurochkin et al. NeurIPS 2019, Figure 4). (ii) BoW is a special case of WMD (Proposition 1 of our paper). Comparing WMD with BoW is especially important in this regard. (iii) One of the main findings of our paper is that even BoW and TF-IDF (weak baselines in Kusner et al.) are sometimes comparable to WMD, and it would be less informative to find that strong baselines are comparable to WMD.
> > >
> > > We've revised the manuscript and mentioned that Kusner et al. compared WMD to many other methods that performed much better than BoW.
> > >
> > >
> > > > The figures 4 and 5 presented in the paper are dependent on the choice of word embeddings
> > >
> > > The WMD paper proposed to use 300-dimensional word2vec, and the following researches have followed this. We also used 300-dimensional word2vec in Figures 4 and 5. Although different word embeddings could be theoretically used, investigating the behavior of WMD with word2vec is of importance in many practical cases.
> > >
> > >
> > > > Again calling the figure 1 misleading is not appropriate in my opinion.
> > >
> > > We've revised it to make the description safer.

---

> > > > ### Comment · Reviewer_id2C · 2021-11-29
> > > > **Thanks for the revision, but my assessment does not change**
> > > >
> > > > I appreciate the response and revision. I indeed hadn't read the code of Kusner et al. It is good to that you pointed out that this bit was missing, but I disagree with the aggressive framing against the original paper. I would like to point out that the figure 2 in the paper under review cites figure 3 in Kusner et al, but it is clearly misrepresenting the original, as it includes only some of the bars of the original (three of the original eight). Also, the WMD paper proposed to use recent results on word embeddings; that was word2vec in 2015, but there has been a lot of progress since.
> > > >
> > > > Overall, I agree with the paper that there are issues in distance metric evaluation that are worth writing a paper on: choice and pre-processing of datasets, lessons that can be learned and transferred between the classic and the modern baselines, see how the results are affected by the choice of embeddings, etc., similar in spirit to the paper by Levy et al. I mentioned earlier. However the paper focuses on criticising one paper, and sometimes unfairly; thus it is not informative.

---

### Official Review · Reviewer_5Jna · 2021-11-02

**Correctness:** 3
**Technical Novelty And Significance:** 4
**Empirical Novelty And Significance:** 3
**Recommendation:** 8
**Confidence:** 4

**Main Review:**

This paper targets a fairer evaluation of WMD. This is a critical problem worth exploring as WMD is a fundamental technique in various research fields. The core claims in this paper include:
1. In the original study of WMD, many duplicate samples exist in the datasets, and applying L2 norm to word embeddings is not explicitly stated.
2. The superiority of WMD over BOW and TF-IDF will weaken enormously by normalizing the BOW and TF-IDF.
3. Both the normalization of word vectors and document-level distance metric (L1 or L2) will impact performance.
4. WMD coincides with L1/L1 BOW empirically and theoretically, consistent with the two modalities characteristic of high-dimensional word embeddings.

By designing extensive experiments, the authors present the above observations and corresponding suggestions. They also provide clean datasets without duplicated samples and related code for further research. Following are some of my questions.

In figure 4, what metric is used for word-level distance?  Experiments show that L1 document-level distance with L1-normalized BOW/TF-IDF will generate the best performance. Regarding WMD, I wonder about the effects of different word-level distances (e.g., L1, L2, or Cosine). For example, WMD uses L2 word-level distance, then what about using other metrics? Besides, what if we use L1 normalization for word embeddings in WMD?

In my experience, the performance of WMD is extensively affected by the removal of appropriate stop-words. This paper only mentions the stop-word strategy on page 7.  I am wondering if the author removed the stop words in other experiments.

Table 3-5 shows that removing OOV words brings significant performance degradation on bbcsports and ohsumed, which is somewhat against expectations. Is there any explanation for this observation?



**Summary Of The Paper:**

This paper empirically shows that the performance of WMD is not as high as initially reported, and the real performance is comparable to  L1-normalized BOW, which can be formulated as a specific case of WMD. The authors also find that WMD resembles BOW in high-dimensional spaces.

**Summary Of The Review:**

This paper re-evaluates the Word Mover's Distance by well-designed experiments. They reveal the true performance of WMD and draw its relationship with L1-normalized BOW. I am more inclined to accept this paper.

---

> ### Author Response · Authors · 2021-11-15
> **Official Author Response**
>
> Thank you for the detailed and positive review. We clarify your concerns in the following.
>
> > In figure 4, what metric is used for word-level distance?
>
> We used L2 distances for the word-level distance in Figure 4 (and throughout our paper).
>
> > Regarding WMD, I wonder about the effects of different word-level distances (e.g., L1, L2, or Cosine).
>
> > what if we use L1 normalization for word embeddings in WMD?
>
> We conducted the experiments. Here are the results.
>
> | | bbcsports | twitter | recipe | ohsumed | classic | reuter | amazon | 20news |
> | --- | --- | --- | --- | --- | --- | --- | --- | --- |
> | WMD {L2 distance / L2 normalized} | 5.1 ± 1.2 | 29.6 ± 1.5 | 42.9 ± 0.8 | 44.5 | 2.9 ± 0.4 | 4.0 | 7.4 ± 0.5 | 26.8 |
> | WMD {L1 distance / L2 normalized} | 4.8 ± 0.9 | 29.6 ± 1.3 | 43.2 ± 1.0 | 44.7 | 2.9 ± 0.3 | 4.1 | 7.3 ± 0.3 | 26.9 |
> | WMD {cosine / L2 normalized} | 5.5 ± 0.8 | 29.4 ± 1.3 | 43.8 ± 0.6 | 45.3 | 2.8 ± 0.4 | 3.4 | 6.6 ± 0.4 | 26.7 |
> | WMD {L2 distance / L1 normalized} | 5.0 ± 1.1 | 29.5 ± 1.4 | 42.9 ± 1.0 | 43.9 | 2.9 ± 0.4 | 4.1 | 7.4 ± 0.5 | 26.6 |
> | WMD {L1 distance / L1 normalized} | 5.1 ± 1.3 | 29.7 ± 1.4 | 43.0 ± 0.8 | 44.1 | 2.9 ± 0.4 | 4.2 | 7.3 ± 0.4 | 26.7 |
> | WMD {cosine / L1 normalized} | 5.8 ± 1.6 | 29.5 ± 1.5 | 43.9 ± 0.9 | 44.8 | 2.7 ± 0.4 | 4.2 | 6.6 ± 0.4 | 26.8 |
>
> The word-level normalization and word-level distances do not affect the results much.
>
> However, notably, {cosine / L2 normalized} performs slightly better in reuter and amazon. Although the differences are subtle, we interpret this result as follows: In bbcsports, twitter, recipe, ohsumed, and 20news, WMD did not perform well in the first place in Table 2. This means that utilizing word similarity does not benefit the performance much on these datasets. Rather, clearly differentiating words (i.e., BoW) is sometimes helpful on these datasets. By contrast, in classic, reuter, and amazon, WMD performed well in Table 2. This means that utilizing word similarity benefits the accuracy on these datasets. The cosine distance magnifies the difference in distances. For example, if the L2 distance is in [0.9, 1.1], the cosine distance is in [0.81, 1.21]. The magnification alleviates the two-modality and helps to utilize the word similarity, and thereby the cosine similarity improves the accuracies on reuter and amazon in the above table.
>
> > I am wondering if the author removed the stop words in other experiments.
>
> We removed the stop words in all experiments except for the Twitter dataset, on which the original paper did not remove the stop words owing to the small documents.
>
> > Table 3-5 shows that removing OOV words brings significant performance degradation on bbcsports and ohsumed, which is somewhat against expectations. Is there any explanation for this observation?
>
> The task of bbcsport is to classify news articles into athletics, cricket, football, rugby, or tennis. In this dataset, many athletes' and groups' names appear. Although these names are strong hints for the class, they are removed as OOV. For example, Hodgson, Lleyton, Henman, Kallis, Harmison, IAAF, and Middlesbrough appear in more than 10 documents in this dataset, but they are removed as OOV. This lowers the performance.
>
> The task of ohsumed is to classify medical abstracts. This dataset contains many medical terminologies, such as Escherichia, lymphoblastic, mRNA, immunoperoxidase, carinii, and Kaposi. Each of these terminologies appears in more than 40 documents in this dataset. Although they are strong hints for the class, they are removed as OOV. This lowers the performance.
>
> ---
>
> We hope that these responses have helped address your concerns. We would appreciate it if you would consider increasing the score and support our paper for acceptance in the discussion.

---

> > ### Comment · Reviewer_5Jna · 2021-11-16
> > **Response to the authors**
> >
> > I appreciate the authors' detailed response, which well addresses my concerns. I will raise my score.

---

### Official Review · Reviewer_M3dN · 2021-11-04

**Correctness:** 3
**Technical Novelty And Significance:** 3
**Empirical Novelty And Significance:** 3
**Recommendation:** 8
**Confidence:** 4

**Main Review:**

# Strengths

This paper identifies issues with the original paper proposing WMD and during the evaluation, it shows the gain mainly due to normalization instead of WMD. It offers detailed analysis and experiments to support its claim.

# Weaknesses

n/a


**Summary Of The Paper:**

This paper re-evaluates WMD and identifies issues with the original paper. It shows that the gain from the original paper is not the product of WMD but the normalization. When the normalization is controlled, WMD performs similarly to baseline. Finally, it shows WMD resembles classic BOW when normalization is controlled.


**Summary Of The Review:**

This paper revisits the original WMD paper and offers a detailed evaluation showing what contributes to the performance gain: normalization instead of WMD.

---

> ### Author Response · Authors · 2021-11-15
> **Official Author Response**
>
> Thank you for the positive review. It encourages us very much. We would appreciate it if you would support our paper for acceptance in the discussion.

---

### Author Response · Authors · 2021-11-15
**Official Author Response**

We thank the reviewers for their reviews and constructive comments.

The main concern of the reviewers seems that our paper is too focused on a single paper (i.e., the original WMD paper of Kusner et al.). However, we claim that our paper has a broad impact beyond a single paper.

As for the misleading point 1, the datasets and the evaluation protocol are used in many papers, including Huan et al. NeurIPS 2016, Yurochkin et al. NeurIPS 2019, Le et al. NeurIPS 2019, Takezawa et al. ICML 2021, Wu et al. EMNLP 2018, Mollaysa et al. ICML 2017, Gupta et al. AAAI 2020, Skianis et al. AISTATS 2020. It harms the credibility of the evaluations if the community continues to use the datasets without noticing the pitfalls. Sharing the issue in the community has substantial importance.

The misleading point 2 indeed confused the authors and readers of the word rotator's distance (Yokoi et al. EMNLP 2020).

As for the misleading point 3, the non-effective normalization methods are used in many following papers, including Yurochkin et al. NeurIPS 2019, Li et al. WWW 2019, Werner et al. 2020, Wrzalik et al. 2019. The importance of reporting the issue is beyond a single paper here as well.

These relevant papers indicate that the impact of our paper is not limited to a single paper.

Investigating the pitfalls of the evaluation protocol may not look sensational as, say, proposing a BERT-style model, but we believe it is important and necessary work for the community.

We also claim that the original WMD is not an out-of-date technique. As we reviewed in the related work section, many WMD-style methods have been proposed in the pre-trained model era, including Yurochkin et al. NeurIPS 2019, Le et al. NeurIPS 2019, Yokoi et al. EMNLP 2020, Takezawa et al. ICML 2021, to name a few. They still thrive in many situations, e.g., when training datasets are not available for fine-tuning and in low-resource devices. Reviewer 5Jna also pointed out that ``WMD is a fundamental technique in various research fields.'' Not a few members in the ICLR community (at least Reviewer M3dN, Reviewer 5Jna, and we) are interested in the re-evaluation of WMD.

---

### Decision · Program_Chairs · 2022-01-20

**Decision:**

Reject

**Comment:**

The authors conduct extensive experiments to show that there were some errors in the original claims of the WMD paper and as opposed to what was claimed in the original paper, WMD does not outperform simpler baselines like BOW and TF-IDF. The authors claim that this is significant because WMD is widely used in the literature and hence pointing out errors in the original paper may help the community.

Out of the 4 reviewers, 1 reviewer wrote a very short review and despite reminders did not elaborate on the reasons for a "Strong Accept". The other reviewer with a "Strong Accept" rating also did not champion the paper in the final discussions. The main objection of the two reviewers who were not in favor of accepting the paper were that (i) it focuses on cirticising a single paper and (ii) some of the criticism is not fair. In response, the authors claim that given the huge amount of derivative work which uses or builds upon the original WMD metric it is crucial to point out these errors.

Having read the reviews and the responses, it is not clear to me whether such a paper which focuses only on such criticism of a single paper (not matter how popular it is) has enough merit in being accepted. Alternatively, if such criticism was a part of a broader work (maybe a work on new document similarity metrics) then it would have more merit. Further, it should be noted that of the 4 misleading conclusions of the original paper identified by the authors at least 2 are debatable (one being an error in the dataset and the other being a normalisation technique which was not mentioned in the paper but used in the code). The authors have also rephrased one of the original 4 misleading points and from the discussion it seems that they agree it is not misleading. It would have been easier for me to accept the paper if it had a new metric and ablation studies which showed that (i) Hey, normalisation is important and should be done for all baseline algorithms that are being compared (ii) Hey, there are errors in the dataset which affect the results